# The Significance of Asymmetry in the Assessment of Brain Perfusion in Atypical Tauopathic Parkinsonian Syndromes

**DOI:** 10.3390/diagnostics12071671

**Published:** 2022-07-09

**Authors:** Piotr Alster, Natalia Madetko, Bartosz Migda, Michał Nieciecki, Michał Kutyłowski, Leszek Królicki, Andrzej Friedman

**Affiliations:** 1Department of Neurology, Medical University of Warsaw, ul. Kondratowicza 8, 03-242 Warsaw, Poland; natalia.madetko@wum.edu.pl (N.M.); andrzej.friedman@wum.edu.pl (A.F.); 2Diagnostic Ultrasound Lab, Department of Pediatric Radiology, Medical Faculty, Medical University of Warsaw, ul. Kondratowicza 8, 03-242 Warsaw, Poland; bartoszmigda@gmail.com; 3Department of Nuclear Medicine, Children’s Memorial Health Institute, al. Dzieci Polskich 20, 04-736 Warsaw, Poland; msnieciecki@gmail.com; 4Department of Radiology, Mazovian Brodno Hospital, ul. Kondratowicza 8, 03-242 Warsaw, Poland; michael.kutylowski@gmail.com; 5Department of Nuclear Medicine, Medical University of Warsaw, ul. Banacha 1a, 02-097 Warsaw, Poland; leszek.krolicki@wum.edu.pl

**Keywords:** corticobasal syndromes, progressive supranuclear palsy, PSP, CBS, single photon emission computed tomography

## Abstract

Progressive supranuclear palsy syndrome (PSPS) and corticobasal syndrome (CBS) are clinical manifestations of tauopathic Parkinsonian syndromes. Due to their overlapping symptomatology, the differential diagnosis of these entities may be difficult when bounded to clinical assessment. The manifestations are commonly associated with pathological entities—corticobasal degeneration and progressive supranuclear palsy, which are four-repeat tauopathies. In this study, the authors attempted to find whether the asymmetry typically associated with CBS may be feasible in the interpretation of perfusion single-photon computed tomography. The analysis based on the examination of patients with progressive supranuclear palsy—Richardson syndrome (PSP-RS), progressive supranuclear palsy—Parkinsonism predominant (PSP-P), and corticobasal syndrome (CBS) revealed significant asymmetry of perfusion of the amygdala in corticobasal syndrome. The more pronounced abnormalities of perfusion were observed in the left amygdala among patients with more severe Parkinsonian syndromes in CBS on the right. This study shows that the comparison of the perfusion of tauopathic Parkinsonian syndromes should be extended by asymmetry analysis. Interestingly, the differentiating potential of brain perfusion is present in the comparison of CBS and PSP-RS, but not in CBS and PSP-P. This phenomenon could be explained by more distinct asymmetry in the perfusion observed in PSP-P, which diminishes the differentiating potential of this parameter when it comes to the comparison of PSP-P and CBS. To the best of our knowledge, this is the first study evaluating which structures can be interpreted as significantly asymmetrical in the context of perfusion in CBS.

## 1. Introduction

Corticobasal syndrome (CBS) is the clinical manifestation of various pathologies [1,2,3]. It is commonly associated with corticobasal degeneration (CBD), a pathological entity that is a four-repeat tauopathy; however, only about half of cases show this correlation [4,5,6]. Other pathologies that are associated with CBS include Alzheimer’s disease (AD), frontotemporal degeneration with transactive response DNA-binding protein 43 kDa (TDP) inclusions, Creutzfeldt–Jakob disease, and progressive supranuclear palsy [7]. CBD is described as a pathology affected by the asymmetry of cortical atrophy, ballooned neurons, nigral atrophy, glial lesions in the gray and white matter, and astrocytic plaques in the cerebral cortex [8]. CBS is a combination of cortical and extrapyramidal syndromes [9]. The contemporary literature presents CBS as more of a group of diseases rather than a single clinical manifestation [3,10]. The heterogeneity of the syndrome is consistent with the recent results concerning the additional examinations in this entity. In a study evaluating Ioflupane ^123^I SPECT (DaTSCAN) in CBS, 44% of the patients did not have any abnormalities. Further evaluation of the cerebrospinal fluid in this group showed an AD cerebrospinal fluid profile in 39% of the CBS patients. The authors associated normal DaTSCAN with possible AD in CBS [11].

Within the evolution of the examination of the cerebrospinal fluid and amyloid biomarkers, CBS was phenotyped as amyloid-positive and amyloid-negative [12]. Amyloid-positive patients were described as being more affected by posterior cortical abnormalities, whereas amyloid-negative patients had more anterior cortical and brain stem dysfunctions. The motor dysfunctions, e.g., Parkinsonism and dystonia, were not more pronounced in any of the subtypes.

The syndrome is commonly associated with an asymmetric manifestation; however, questions may rise when the clinical asymmetry is not as evident and when the supplementary examinations do not reveal any specific findings. The majority of patients diagnosed with CBS are additionally examined using neuroimaging. Among these patients, magnetic resonance imaging (MRI) is the most commonly used method, as it provides the highest spatial resolution of the cerebral structures and enables the most precise evaluation of the morphological changes, such as atrophy. Degeneration of the brain can also be assessed to a certain degree using computed tomography (CT), but as the quality of the visualization of anatomical details is significantly poorer compared to MRI, CT plays a minor role in the diagnostic imaging of these entities. Similarly, less attention is paid to transcranial sonography or nuclear radiotracers. The more specific radiotracers are dedicated to detecting tau. Among them could be mentioned 18F-flortaucipir (AV-1451), which is affected by high cost, low accessibility and off-binding properties [7]. The radiotracer was found to be linked with neuropathologically impacted regions in CBD and PSP. Moreover, the absence of asymmetry in the radiotracer’s accumulation was found in CBS lacking tauopathy. The second generation tau radiotracer [^18^ F]PI-2620 was found to monitor disease progression. The differences in the accumulation of the radiotracer were correlated with the laterality of the disease’s severity [13]. The majority of the work concerning [^18^F]PI-2620 is based on the examination of AD. Most of the research evaluating atypical Parkinsonism is affected by low numbers of examined patients and require further analysis.

The methods that are accessible in clinical practice, such as 18-fluorodeoxyglucose positron emission tomography or hexamethylpropyleneamine oxime (HMPAO) in single-photon emission computed tomography (SPECT), provide an overview on brain metabolism or perfusion [4,14,15]. The earlier findings compared the perfusions of PSPS and CBS and did not provide any significant differences between the entities. None of the studies concentrated on the significance of the differences between the bilateral structures and possible regions of interest (ROI) showing more pronounced asymmetry. In this study, the authors intended to verify whether the evaluation of the asymmetry of the perfusion of specified bilateral ROI could be feasible in the differentiation of tauopathic Parkinsonian syndromes. An earlier study concerning the role of SPECT in the evaluation of atypical Parkinsonism showed reduced perfusion in the frontal lobe in PSP and asymmetrical perfusion of the frontal, parietal, and temporal lobes in CBD [16,17]. In this work, the authors did not distinguish subtypes of PSP; moreover, the work was based on the evaluation of four cases [16,17].

The presented features combined with CBS overlapping with progressive supranuclear palsy (PSPS) leads to a necessity to search for efficient methods for additional examinations [4,5]. Currently, the only method enabling a definite diagnosis of CBD is neuropathological examination; however, the examination of CBS, due to various clinical manifestations, seems bounded by obstacles [3].

## 2. Material

A total of 54 patients with tauopathic Parkinsonian syndromes were included in this study, among which 21 were diagnosed with PSP-Richardson syndrome (PSP-RS), 14 with PSP-Parkinsonism predominant (PSP-P), and 19 with CBS. The patients were recruited between January 2017 and December 2019. The groups were age-matched (Table 1). The diagnoses were based on the contemporary criteria [18,19]. All of the patients with PSP-P and PSP-RS had symmetrical clinical manifestations. Patients with CBS had asymmetrical clinical manifestations. All of the patients were assessed and diagnosed by neurologists experienced in movement disorders at the Department of Neurology of the Medical University of Warsaw. All of the patients included in this study were hospitalized at the Department of Neurology. Each patient signed a written consent form before initiating the study. Patients with coexistent cerebrovascular diseases and with neoplasms and other types of focal brain damage were excluded from the study.

## 3. Methods

### 3.1. SPECT

The examination using SPECT was performed at the Department of Nuclear Medicine at Mazovian Brodno Hospital. The assessment of perfusion was performed using the same method as previous studies [5]. The examination of cerebral blood flow was conducted using technetium-99m hexamethylpropyleneamine oxime ([^99m^Tc]Tc-HMPAO). A quantity of 740 mBq of [^99m^Tc]Tc-HMPAO was dispensed to the patients in a quiet, dimly lit room. The acquisition was performed in a supine position with a SPECT/CT scan (Symbia T6, Siemens) on a dual-head gamma camera with a low-energy high-resolution parallel-hole collimator. A step-and-shoot acquisition mode was utilized. Sequences of 128 frames on a 128 × 128 matrix were used (64 projections per head, 30 s per projection). The photopeak was set at 140 keV with a 10% window on either side of it. Iterative reconstruction (eight iterations, eight subsets, 7 mm Gauss filter), scatter correction, and CT attenuation correction were performed. Post-processing examination was carried out using Scenium software (Siemens Medical Solutions USA, Inc., Malvern, PA, USA). The SPECT ROIs were preplanned using Scenium software (an integral part of the Siemens workstation) based on T1-weighted MRI images of a standard brain dataset. The shape and size of the SPECT-examined brains were calibrated according to the shape and size of the standard brains from the dataset. The pre-planned ROIs were then extrapolated to the SPECT images of the assessed brains. Eventually, total maximum and minimum counts were automatically evaluated in each ROI of the investigated brain SPECT scans and were differentiated using Scenium with measurements from the standard brain SPECT scan datasets. No control group was additionally assessed in this study. The data were compared to a reference database comprising the [^99m^Tc]Tc-HMPAO brain scans of 20 healthy volunteers with an age range of 64–86 years old (males and females). All comparisons were automatically shown as standard deviations using Scenium. The values of standard deviations from ROIs were examined in multiple locations in the brain by statistical analysis. The results of the examination were evaluated by a specialist experienced in nuclear medicine.

### 3.2. Statistical Analysis

All calculations were performed using Statistica software (version 13.1 Dell. Inc. Statsoft, Round Rock, TX, USA). For the analysis, we used the computed delta parameters (Δ) expressed as the absolute values of the differences of the SPECT perfusions between the right and left sides in each analyzed region. Data distribution was assessed with the Shapiro–Wilk test. Due to non-normal distribution, all parameters are expressed as medians with their lower (Q1) and upper (Q3) quartiles and their interquartile ranges (Q1–Q3). For group comparisons, we used Kruskal–Wallis ANOVA. For post hoc analysis, we used the pairwise multiple comparison of mean ranks (PMCMR). Significant results are presented as box plots. In terms of multiple comparison correction, we used the Bonferroni correction to control the false discovery rate (FDR). The calculated *p* value of 0.00625 was considered significant.

## 4. Results

The median values within the interquartile range (Q1–Q3) of the assessed parameters for the whole group and in the subgroups are listed in Table 1. The highest values of the absolute SPECT perfusion differences between the right and left sides of the amygdala, the basal ganglia, the cerebellum, and the thalamus were found in CBS patients, Table 1. In the frontal lobe and the hippocampus, we did not observe any statistically important differences between the median values of delta parameters in all of the groups—patients with PSP-RS, PSP-P, and CBS, Table 2. Those values were equal between groups (hippocampus) or nearly equal (frontal lobe), Table 1. The insular and temporal lobe patients with PSP-RS and CBS tended to have approximately the same value of absolute SPECT perfusion difference between the left and right sides, whereas patients with PSP-P presented the highest or lowest Δ values depending on the region: for the insular lobe, it was the highest value (Δ Insular = 2.1), for the temporal lobe, it was the lowest value (Δ temporal lobe = 0.5). When we compared the Δ SPECT perfusion values the between analyzed groups, we found that those groups differed between each other in only the amygdala region according to the corrected *p* value, *p* = 0.0015, Table 2. In addition, we believe that two examined regions in which we noticed low *p* values (lower than 0.05 but higher than the *p* value with the Bonferroni correction of 0.00625) should be mentioned: the basal ganglia (*p* = 0.0235) and the temporal lobe (*p* = 0.0083), as seen in Table 2. Further post hoc analysis with the pairwise multiple comparison of mean ranks revealed that in the amygdala region, statistically important differences of absolute SPECT perfusion between the right and left sides were found in patients with CBS and PSP-RS, *p* = 0.0009. The Δ SPECT parameters were CBS = 1.2 and PSP-RS = 0.6. There was no significant difference in the Δ SPECT parameters between PSP-RS vs. PSP-P and CBS vs. PSP-P: *p* = 0.2388 and 0.4583, respectively, as shown in Table 3 and in Figure 1.

## 5. Discussion

The novelty of this research is related to the significance of the amygdala in the examination of tauopathic Parkinsonian syndromes (Figure 2). Moreover, the results suggest that evaluating the asymmetry should be based on specifically predefined regions of interest. The asymmetry of cerebral blood flow (CBF) was evaluated in a work examining PD and CBS using arterial spin labeling magnetic resonance imaging. The authors indicated greater asymmetry of CBF in the perirolandic area and in the parietal cortex in CBS when compared to PD. Moreover, it was found that CBF asymmetry has more pronounced sensitivity than volume asymmetry [20]. The issue concerning the abnormalities in neuroimaging in CBS has been described in a few papers. One of them presented volume loss in the putamen, the hippocampus, the nucleus accumbens bilaterally and in the corpus callosum and the right amygdala in T1-weighted 3Tesla MRI [21]. In a recent study conducted on 15 CBS patients and 14 PSP patients who were examined using [^11^C]C-UCB-J and a synaptic vesicle glycoprotein 2A radiotracer, reduced accumulation was found in the frontal, temporal, parietal, and occipital lobes; in the cingulate cortex, the hippocampus, the insular cortex, and the amygdala; and in the subcortical structures. The reduction was observed in CBS and PSP [22]. The radiotracer can be interpreted as an indicator of synaptic density. In a work evaluating speech and language functions in PSP, CBS, and a non-fluent variant of primary progressive aphasia, it was also found that the volume of the amygdala was associated with Mini Linguistic State Examination. [23]. In a study describing a CBD patient surviving 18 years after diagnosis, it was found that in the neuropathological evaluation, an abnormal accumulation of TDP-43 and alpha synuclein was detected in various regions throughout the brain beyond the amygdala and other limbic areas [24]. Another study evaluating ubiquitin-positive achromatic neurons in CBD revealed that the amygdala is among the regions that are highly affected by increased densities of these neurons [25]. A paper on memory deficits in a demented patient with CBD revealed that atrophic changes were present in the frontal, parietal, and central areas, while the amygdala, the hippocampi, and the temporal lobes were relatively spared [26]. A different study stated that the amygdala and the hippocampus may be vulnerable to tau and alpha-synuclein pathologies [27]. The differences between CBD-CBS and CBS based on Alzheimer’s disease pathology revealed that CBD-CBS patients are more likely to have posterior frontal atrophy, while AD-CBS patients have more disseminated frontal and parietal atrophy [28]. Regardless of the features associated with CBD and AD pathologies, the observations concerning the atrophies in the frontal and parietal regions should be considered more as tendencies than clear tools providing a definite assessment.

According to the contemporary criteria of diagnosis, asymmetry is among the features of CBS. Recently, it has also been described in the “most common symmetrical” atypical Parkinsonism—PSP. The laterality index in PSP-RS did not significantly differ when compared to PSP-parkinsonism predominant (PSP-P) and PSP-corticobasal syndrome (PSP-CBS). The authors revealed that Parkinsonism, dystonia, and myoclonus may be asymmetrically present in 53.6%, 21.4%, and 17.9% of cases of PSP-RS, respectively [29]. Interestingly, the commonly discussed asymmetry of CBD is not always evident. In a series of five cases, which were neuropathologically confirmed with CBD, the clinical manifestation was symmetrical. Features such as apraxia, myoclonus, dystonia, and alien limb phenomenon were not observed [30]. The asymmetry is most commonly revealed in MRI of CBS [21]. Due to the overlaps in the clinical manifestation and possibly in neuroimaging, the examination of differences between bilateral structures should be verified by more restrictive thresholds of significance. Less attention has been paid to the asymmetry of selected structures in CBS.

The clinical differentiation of CBS and PSP-RS seems less evident than between CBS and PSP-P. The studies concentrating on the differences between CBS and PSP-RS showed higher width of the third ventricle in CT and in other methods of neuroimaging [31,32]. Additionally, certain works showed the hyperechogenecity of substantia nigra in CBS in transcranial sonography [33]. Generally, the differences between PSP-RS and CBS were not found to be pronounced. In, Previous studies concerning the differences between PSP-RS and CBS in terms of [^99m^Tc ]Tc-HMPAO, SPECT did not show any significant differences in the comparison of brain perfusion [5]. The previous examination of perfusion in SPECT did not present the PSP phenotypes separately. Moreover, the differences between bilateral structures were not measured. Other studies showed that the differentiation of PSP and CBS may be jeopardized due to the diversity of underlying pathologies, especially in the context of CBS. Likely due to the significant clinical differences between PSP-P and CBS, to the best of our knowledge, no studies dedicated to the differentiation of CBS and PSP-P have been performed. We hypothesize that our results indicating no differentiating potential in the comparison of perfusion asymmetry between CBS and PSP-P could be explained by the diseases’ clinical course and symptomatology. In its early stages, PSP-P resembles PD, which is known for its asymmetry in terms of clinical its manifestation. Therefore, as both CBS and PSP-P may manifest with asymmetric symptomatology, differences in bilateral perfusion are not differentiating. In the examined patients with PSP-P, no asymmetrical clinical manifestation was observed at the moment of examination; however, the possible asymmetry may be associated with earlier stages. This phenomenon implicates the issue of a possible continuum of neurodegeneration in Parkinsonian syndromes, which requires further research.

The majority of works concerning CBS are based on a series of cases or a small number of patients, which additionally makes interpreting the results difficult. This study should be interpreted as a factor in the ongoing discussion concerning the actual boundaries between clinical manifestations of tauopathic Parkinsonian syndromes. On one hand, the differences regarding the asymmetry of perfusion of certain ROIs suggest a differentiating feature between PSP-RS and CBS. On the other hand, the diversity of the underlying pathologies of CBS may make the examination and interpretation of neuroimaging difficult. The overlaps in the clinical manifestation of PSP-RS and CBS may be interpreted as a consequence of combined pathomechanisms of related tauopathies [34,35,36] The interpretation of diverse pathways leading to the diseases becomes even more striking when associating them with the recently described vascular PSP syndrome and CBS [37]. CBS has also been also described as a possible manifestation of hypoparathyroidism [38]. This may suggest that PSPS and CBS are more results of mechanisms possibly related to microglial activation or oxidative stress rather than the direct manifestations of pathologies [39,40,41] Possible extended examinations of the amygdala combined with neuropsychological evaluation could result in an overview of the significance of the possible emotional component of CBS. The results of the assessments of PSP-P and PSP-RS suggest that PSP-P should not be classified as a more benign form of the evolution of the disease, but rather a more complex clinical course.

## 6. Limitations

The study is affected by several limitations. Instead of a control group, the authors referred to the measurements of 20 healthy volunteers from the software databases. The groups examined in the work were relatively small; however, the authors intended to verify rare diseases. Moreover, in the context of PSP patients, they were additionally divided into two subgroups. The CBS group consisted of a significantly higher number of females; however, the authors did not find works concerning the impact of sex on perfusion in CBS. No additional neuroimaging was evaluated in this study. Due to the fact that the examined patients are alive, the diagnosis was based on clinical examinations and was not extended by neuropathological assessment; however, the goal of this research was to obtain tools facilitating in vivo examination, which could be feasible after the introduction of disease-modifying treatments. CBS as well as the other syndromes evaluated in the study are likely manifestations of various pathologies, which may impact the results.

## 7. Conclusions

This study stresses that the examination of CBS should be evaluated using the asymmetries of perfusion of precisely indicated bilateral structures. The differentiative potential of perfusion differences between the left and right amygdala indicated a more precise perspective of examination, but may also suggest links between CBS’ clinical manifestation and emotional presentation. Previous studies concerning the amygdala in CBS do not provide an unequivocal perspective on the examination of atrophies in CBS. Little is known about the role of the amygdala in CBS. This may be related to the fact that the conclusions of this study are based on clinically diagnosed CBS, which is not necessarily associated with CBD. This may be interpreted as a possible explanation of the discrepancies in the previous results of CBD and CBS works [10]. The differentiative potential of the amygdala is only between CBS and PSP-RS, which suggests the necessity of neuropsychological and speech therapist evaluation. The significance of AD’s pathology in CBS may be crucial in future therapy. Future studies based on larger groups of patients should provide an overview in this area.

## Figures and Tables

**Figure 1 diagnostics-12-01671-f001:**
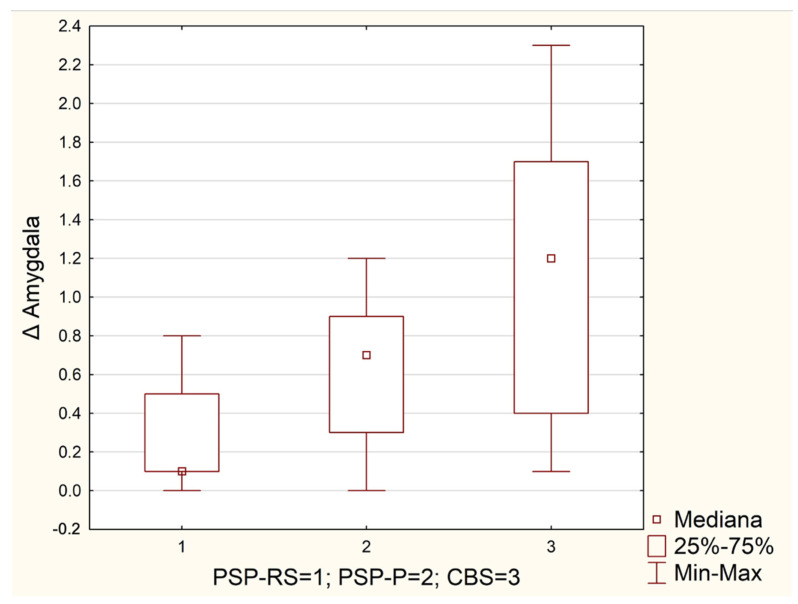
Δ Amygdala in tauopathic Parkinsonian syndromes.

**Figure 2 diagnostics-12-01671-f002:**
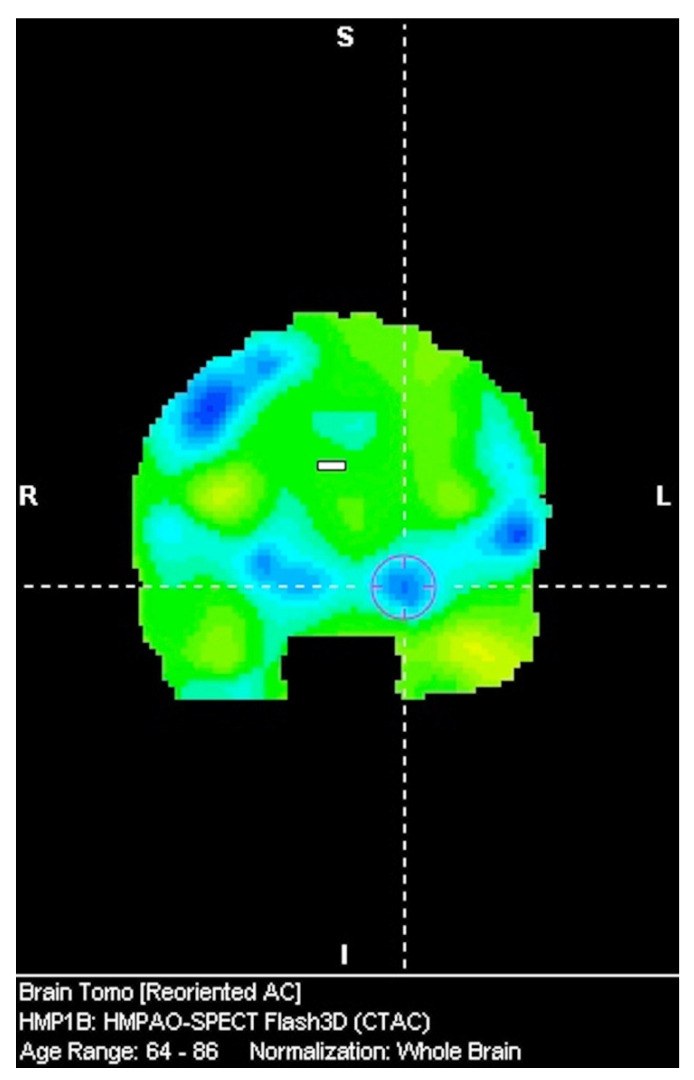
Hypoperfusion of the amygdala in CBS.

**Table 1 diagnostics-12-01671-t001:** Descriptive statistics.

Parameter	Whole Group (N = 54)				PSP-RS (N = 21)				PSP-P (N = 14)				CBS (N = 19)			
	Median	Lower Quartile	Upper Quartile	Quartile Range	Median	Lower Quartile	Upper Quartile	Quartile Range	Median	Lower Quartile	Upper Quartile	Quartile Range	Median	Lower Quartile	Upper Quartile	Quartile Range
Age	72.5	69	77.5	8.5	72	70	78	8	73	64	77	13	73	72	75	3
Δ Amygdala	0.6	0.2	1.2	1	0.1	0.1	0.5	0.4	0.7	0.3	0.9	0.6	1.2	0.4	1.7	1.3
Δ Basal Ganglia	0.7	0.35	1.1	0.75	0.6	0.3	1	0.7	0.4	0.1	0.85	0.75	0.8	0.7	1.3	0.6
Δ Cerebellum	1	0.7	1.6	0.9	1	0.7	1.4	0.7	0.85	0.6	1.4	0.8	1.5	0.8	2.6	1.8
Δ Frontal Lobe	0.6	0.4	1.4	1	0.6	0.2	1.5	1.3	0.7	0.5	0.9	0.4	0.6	0.4	1.2	0.8
Δ Hippocampus	0.7	0.4	1.05	0.65	0.7	0.4	1.1	0.7	0.7	0.55	0.85	0.3	0.7	0.3	1.1	0.8
Δ Insula	1.6	0.9	2.4	1.5	1.6	1.1	2.5	1.4	2.1	0.9	2.2	1.3	1.5	0.2	2.4	2.2
Δ Temporal Lobe	1.4	0.6	2.75	2.15	1.7	0.7	2.8	2.1	0.5	0.4	0.7	0.3	1.7	1.1	3.4	2.3
Δ Thalamus	0.7	0.2	1.4	1.2	0.6	0.25	1.3	1.05	0.65	0.5	1.2	0.7	0.7	0.1	1.6	1.5

**Table 2 diagnostics-12-01671-t002:** Descriptive statistics.

Parameter	Kruskal–Wallis ANOVA
	*p*
Δ Amygdala	0.00
Δ Basal Ganglia	0.02
Δ Cerebellum	0.10
Δ Frontal Lobe	0.84
Δ Hippocampus	0.99
Δ Insula	0.32
Δ Temporal Lobe	0.01
Δ Thalamus	0.90

Legend: statistically significant *p*-values are in red.

**Table 3 diagnostics-12-01671-t003:** Post hoc analysis.

Parameter		*p*		
		PSP-RS	PSP-P	CBS
Δ Amygdala	PSP-RS		0.24	0.00
	PSP-P	0.24		0.46
	CBS	0.00	0.46	

Legend: statistically significant *p* values for the pair wise multiple comparisons of mean ranks are in red.

## Data Availability

The data is available on request.

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
