# Peer review of "The Significance of Asymmetry in the Assessment of Brain Perfusion in Atypical Tauopathic Parkinsonian Syndromes"

_diagnostics, 2022, doi:10.3390/diagnostics12071671_

Round 1

Reviewer 1 Report

In this study authors attempted to find whether the asymmetry typically associated with corticobasal syndrome (CBS) may be feasible in the interpretation of perfusion single photon computed tomography. They found that the differentiating potential of brain perfusion was present in the comparison of CBS with PSP-Richardson syndrome (RS), not in CBS and PSP-Parkinsonian subtype (PSP-P). 

The manuscript is of interest, the methods and results are sound. I have some comments:

- Title: I would specify that these conditions are tauopathic atypical parkinsonian disorders.

- Abstract: I would specify here and in the main text that these disorders are 4R (four-repeat) tauopathies.

- Introduction: I would underline here that CBD is a pathological entity.

- Material: Please, avoid specifying sex of males and females per each category; the author can specify in brackets the number of males or of females consistently per each category.

- Methods: Page 4, line 4, before "The acquisition" the authors missed a period.

- Discussion, page 8: I would change "beneficial form of evolution" with "benign disease entity" or something similar.

- Tables: Please, limit the digits to 2 after the period (e.g., 0.00). Also, Table 3 legend has different fonts.

Author Response

Dear Reviewer,

We would like to thank for all the valuable comments and suggestion. We have revised the manuscript accordingly: 

"Title: I would specify that these conditions are tauopathic atypical parkinsonian disorders.

- Abstract: I would specify here and in the main text that these disorders are 4R (four-repeat) tauopathies.

- Introduction: I would underline here that CBD is a pathological entity.

- Material: Please, avoid specifying sex of males and females per each category; the author can specify in brackets the number of males or of females consistently per each category.

- Methods: Page 4, line 4, before "The acquisition" the authors missed a period.

- Discussion, page 8: I would change "beneficial form of evolution" with "benign disease entity" or something similar.

- Tables: Please, limit the digits to 2 after the period (e.g., 0.00). Also, Table 3 legend has different fonts."

All of the suggestions above were implemented in the manuscript.

Best regards 

Piotr Alster

Reviewer 2 Report

v                 Suggestions to Author/s

  1. Dear Piotr Alster, as a selected reviewer I made the prompt check of your excellent article:” The significance of asymmetry in the assessment of brain perfusion in tauopathic parkinsonian syndromes” and found it: (X) Excellent, accept the submission (5).

2.      In the text, some minor mistakes were found. In the .doc form of text, they are labeled  with a red color. Please make them black again and convert text from .doc to PDF.

Author Response

Dear Reviewer,

We would like to thank for all of the valuable comments and suggestions. We have revised the manuscript and implemented all of the changes.

Best regards,

Piotr Alster